

# Gluten-free food database: the nutritional quality and cost of packaged gluten-free foods

Benjamin Missbach[1], Lukas Schwingshackl[2], Alina Billmann[1], Aleksandra Mystek[1], Melanie Hickelsberger[1], Gregor Bauer[3] and Jürgen König[1]

[1] Department of Nutritional Sciences, University of Vienna, Vienna, Austria
[2] Department of Epidemiology, German Institute of Human Nutrition, Nuthetal, Germany
[3] Department of Internal Medicine I, Kaiser-Franz-Josef Spital, Vienna, Austria

## ABSTRACT

Notwithstanding a growth in popularity and consumption of gluten-free (GF) food products, there is a lack of substantiated analysis of the nutritional quality compared with their gluten-containing counterparts. To put GF foods into proper perspective both for those who need it (patients with celiac disease) and for those who do not, we provide contemporary data about cost and nutritional quality of GF food products. The objective of this study is to develop a food composition database for seven discretionary food categories of packaged GF products. Nutrient composition, nutritional information and cost of foods from 63 GF and 126 gluten-containing counterparts were systematically obtained from 12 different Austrian supermarkets. The nutrition composition (macro and micronutrients) was analyzed by using two nutrient composition databases in a stepwise approximation process. A total of 63 packaged GF foods were included in the analysis representing a broad spectrum of different GF categories (flour/bake mix, bread and bakery products, pasta and cereal-based food, cereals, cookies and cakes, snacks and convenience food). Our results show that the protein content of GF products is >2 fold lower across 57% of all food categories. In 65% of all GF foods, low sodium content was observed (defined as <120 mg/100 g). Across all GF products, 19% can be classified as source high in fiber (defined as >6g/100 g). On average, GF foods were substantially higher in cost, ranging from +205% (cereals) to +267% (bread and bakery products) compared to similar gluten-containing products. In conclusion, our results indicate that for GF foods no predominant health benefits are indicated; in fact, some critical nutrients must be considered when being on a GF diet. For individuals with celiac disease, the GF database provides a helpful tool to identify the food composition of their medical diet. For healthy consumers, replacing gluten-containing products with GF foods is aligned with substantial cost differences but GF foods do not provide additional health benefits from a nutritional perspective.

Corresponding author
Benjamin Missbach,
benjamin.missbach@univie.ac.at

## INTRODUCTION

Individuals with celiac disease (CD) show high levels of intestinal inflammation when exposed to gluten-containing foods (*Ludvigsson et al., 2013*; *Rubio-Tapia et al., 2013*). In western countries, the prevalence for CD is estimated at approximately 1% (*Golley et al., 2015*; *Gujral, Freeman & Thomson, 2012*). Clinically, as direct response to gluten and related prolamines in a diet, immunological processes damage intestinal mucosa and lead to villous atrophy, crypt hyperplasia and nutrient malabsorption (*Dickson, Streutker & Chetty, 2006*; *Husby et al., 2012*). To get full remission of the symptoms, excluding gluten-containing cereals (e.g. wheat, rye, barley) in a strict lifelong gluten-free (GF) diet is indicated (*El-Chammas & Danner, 2011*; *Green, 2009*). The nutritional quality of GF products that replace cereal-based foods is pivotal for patients with CD. Previous research showed that GF food products differ in their nutrient content compared to gluten-containing counterparts (*Mazzeo et al., 2015*; *Miranda et al., 2014*). In addition, a recent evaluation of more than 600 GF foods in Australian supermarkets showed that it is unlikely that GF foods have health benefits for individuals without CD, in particular due to the reported lower protein content in GF compared to non-GF products (*Wu et al., 2015*). However, contemporary data reporting the nutritional quality is scarce (*Staudacher & Gibson, 2015*), although the popularity of GF food products is increasing among consumers (*Marketsandmarkets.com, 2013*). To the best of our knowledge, no database with respect to nutritional quality for packaged GF food is available in German-speaking countries.

For CD patients, adhering to a restrictive GF diet can be challenging for several reasons. First, food choices are essentially limited because cereal products are staple foods in western countries and play a predominant role in a regular diet (e.g., bread or pasta). Second, a wide range of processed foods contain gluten-based products as additional ingredients. Prior to consumption of these foods, a detailed examination of the ingredient list has to be performed to avoid being exposed to gluten. This requires fundamental nutritional knowledge and a high level of self-discipline (*Mulder et al., 2015*). Third, 20–38% of patients with CD have some nutritional deficiencies due to their medical condition (*Kinsey, Burden & Bannerman, 2008*; *Saturni, Ferretti & Bacchetti, 2010*); e.g iron deficiency, deficiency in B vitamins ($B_6$, $B_{12}$) and trace minerals (e.g., zinc) (*Harris et al., 2012*; *Theethira & Dennis, 2015*). In a nutshell, patients with CD are need to structure their diet in a strict manner to maintain a positive long-term health outcome. Therefore, GF products have been developed as alternatives to cereal-based formulations. A wide range of products based on teff, amaranth, buckwheat or quinoa are now available for consumers exploring different alternatives to enhance sensory properties and shelf-life (*Gallagher, Gormley & Arendt, 2004*; *Pellegrini & Agostoni, 2015*).

Likewise, GF products are very popular among consumers without CD, which has led to almost exponential rise in sales for GF products over the last decade (*Marketsandmarkets.com, 2013*; *Strom, 2014*). *Mardini, Westgate & Grigorian (2015)* report updated US data from the National Health and Nutrition Examination Survey (NHANES) from 2009–2012. From all study participants (14.701 participants), 0.9% adhered to a GF diet, even though

85% of this group was never diagnosed with CD and 99% had negative serological markers for CD. For the majority of consumers, GF products are perceived as healthier than conventional products (*Marcason, 2011*). While evidence for this assumption is not based on solid data (*De Giorgio, Volta & Gibson, 2015*; *Gaesser & Angadi, 2012*), food companies continue to market GF foods as healthier and charge premium prices for their products (*Singh & Whelan, 2011*; *Stevens & Rashid, 2008*). Still, there is no solid data putting GF products into proper perspective about nutritional quality and product costs.

To provide better consumer information, the present work is the first attempt to build a nutrient composition database for packaged GF products available in one German-speaking country (Austria). The aim of the current study is to present data from GF foods representing the main sources of cereal-based food and analyze their nutrient content and cost.

## MATERIALS & METHODS

We used a matched food sample procedure to analyze the nutrient content of packaged GF foods available on the Austrian Market. We grouped packaged GF foods with matching gluten-containing foods from two nutrition databases, estimated their nutrient content by using a step-by-step estimation process and compared the nutritional quality. We defined primary (macronutrient and energy content) and secondary outcome parameter (micronutrients and cost of the products) for further analysis.

### Food products included

Between fall 2014 and spring 2015, packaged GF foods from 19 brands were obtained from three main supermarket chains in Austria in 12 different branches across Vienna. We selected 162 packaged GF foods from seven different food categories representing the majority of consumed processed foods that are originally based on cereal formulations for celiac patients (*Martin et al., 2013*). We only analyzed packaged GF foods marked with the European gluten-free label (*Commission of the European Communities, 2009*), excluded foods with non-verified gluten-free labels and excluded foods from categories that are not gluten-based in their original formulation. Following food categories were analyzed: flour/bake mix, bread and bakery products, pasta and cereal-based food, cereals, cookies and cakes, snacks and convenience food (for detailed listing, see Table S1). Additionally, we assessed the cost for each product.

Both food quality as well as cost ranged widely within individual food products. To minimize this within-product range and provide more homogenous data in both target variables (nutritional quality; product cost), we matched two gluten-containing foods differing in cost range (one budget and one pricier product) for each GF food. From originally 162 identified packaged GF foods, we excluded duplicates (86) and foods with incomplete nutrient information (13 foods) (see flow diagram and detailed list of exclusion in Supplemental Information 1). Our final sample consists of 63 GF and 126 similar gluten-containing foods for subsequent nutrient content matching procedure.

### Step-by-step estimation process for nutrient content

The selected GF foods were matched with two similar gluten-containing foods available in two different databases used in the Austrian Nutrition Surveys (BLS 3.02 Max Rubner Institute, Germany; Austrian Nutrient Database: ÖNWT, dato denkwerkzeuge, Vienna, Austria). We used a Microsoft Excel worksheet to compile the composition in macro- and micronutrients of the GF foods per 100 g in its raw form. We imputed the quantity for each ingredient in a descending order. In a second step, we estimated the quantity of each ingredient for every product based on the percentage of the final recipe and its rank order reported on the label (theoretical nutrient composition). Furthermore we compared the theoretical macronutrient composition of the food with the given information on the food label. The process was reiterated by adjusting the percentage of the different ingredients until the final results reflected the values of energy content reported on the food label. To assess the precision of this procedure, we calculated the estimation precision (theoretical nutrient content/nutrient content on the food label in %). The precision for the estimation of all macronutrients was very good and within an overall variation range of 7% (for detailed listing, see Supplemental Information 2).

As a result of this process, we could estimate the amount of ingredients available in the GF products and extrapolate the nutritional components for following ingredients and nutrients: water (g/100 g), sugar (g/100 g), energy content (in kcal/100 g), macronutrients (carbohydrates, proteins, total fat, saturated fatty acids, monounsaturated fatty acids (MUFA), polyunsaturated fatty acids (PUFA), fibre; all in g/100 g), cholesterol (mg/100 g), minerals (i.e., Iron, Calcium, Sodium, Potassium, Phosphorous and Zinc; all in mg/100 g) and vitamins (Vitamin E, Thiamin, Riboflavin, Niacin, and Vitamin C; all in mg/100 g; Vitamin D, Retinol, $\beta$-carotene equivalents in µg/100 g).

### Statistical analyses and availability of the database

Statistical analyses were conducted using IBM SPSS Statistics 22. Unpaired $t$-test was used to compare means; bivariate comparisons were tested by $\chi^2$ test. The Bonferroni post-hoc test was used to correct for multiple comparisons; $p$-values $< 0.05$ were classified as significant. Post-hoc power analysis was calculated by the difference between two independent means with G*Power 3.1.9 (*Erdfelder et al., 2009*).

The Gluten-Free Food Database (Austria) can be accessed via the science collaboration platform: Open Science Framework. Contributions to the dataset can be made upon request and registration via the online platform (*Open Science Framework*).

## RESULTS

The database provides quantitative information of macro- and micronutrients of the GF product. It contains nutrient data present in the traditional databases of gluten-containing foods (see Tables 1 and 2).

### Primary outcome parameter: macronutrient and energy content

Across all food categories, energy content ranged between 270.5 $\pm$ 13.5 kcal/100 g (category: bread and bakery products) to 398.8 $\pm$ 25.4 kcal/100 g (category: snacks).

**Table 1** Macronutrient composition of gluten-free products in Austria.

| | Energy (kcal) | Protein (g) | Carbohydrates (g) | Sugar (g) | Total fat (g) | Saturated fatty acids (g) | MUFA (mg) | PUFA (mg) | Fiber (g) |
|---|---|---|---|---|---|---|---|---|---|
| **Flour/bake mix** | | | | | | | | | |
| Flour[a] | 345.5 | 3.7 | 77.6 | 1.0 | 1.7 | 0.2 | 0.3 | 1.0 | 3.8 |
| Bake mix white (cake) | 338.4 | 3.2 | 77.5 | 17.8 | 1.2 | 0.3 | 0.5 | 0.5 | 2.3 |
| Bake mix brown (cake) | 394.7 | 2.6 | 82.9 | 52.5 | 5.4 | 3.1 | 1.8 | 0.4 | 2.8 |
| Bake mix (Pizza)[a] | 322.7 | 5.9 | 70.3 | 1.5 | 1.9 | 0.4 | 0.8 | 0.7 | 3.9 |
| Breadcrumbs[a] | 350.8 | 5.9 | 70.1 | 0.4 | 4.8 | 1.3 | 1.9 | 1.2 | 6.0 |
| **Bread/Bakery products** | | | | | | | | | |
| Rustic bread | 238.4 | 3.7 | 51.0 | 0.7 | 1.9 | 0.3 | 0.8 | 0.6 | 1.4 |
| Whole-grain bread | 263.6 | 8.5 | 40.8 | 5.1 | 7.2 | 1.0 | 2.2 | 3.9 | 8.4 |
| Toast[a] | 224.8 | 4.8 | 39.0 | 2.1 | 5.3 | 1.4 | 2.0 | 1.5 | 6.3 |
| Bun | 259.2 | 1.4 | 48.7 | 5.2 | 6.2 | 3.0 | 1.6 | 1.2 | 1.6 |
| Ciabatta | 213.4 | 3.3 | 44.9 | 2.9 | 2.0 | 0.3 | 2.3 | 0.7 | 8.3 |
| Raisin bread | 261.0 | 4.0 | 49.1 | 18.6 | 4.2 | 1.5 | 0.8 | 0.8 | 1.6 |
| Scone | 293.2 | 3.7 | 52.2 | 13.8 | 7.5 | 2.2 | 1.4 | 1.1 | 2.7 |
| Baguette | 270.1 | 4.5 | 56.7 | 6.4 | 2.5 | 0.4 | 1.9 | 3.0 | 5.3 |
| Lye Pretzel | 343.8 | 4.8 | 59.2 | 7.8 | 9.4 | 4.7 | 1.0 | 0.9 | 2.2 |
| Rusk | 343.9 | 0.3 | 82.5 | 0.8 | 0.9 | 0.3 | 3.2 | 0.7 | 0.7 |
| Crispbread | 351.6 | 6.9 | 77.8 | 6.0 | 0.9 | 0.2 | 0.4 | 0.2 | 2.9 |
| Wraps | 228.7 | 3.1 | 38.8 | 0.4 | 7.5 | 2.2 | 0.3 | 0.4 | 2.8 |
| **Pasta and cereal-based products** | | | | | | | | | |
| Fusilli | 335.9 | 8.2 | 69.9 | 1.0 | 2.2 | 0.3 | 0.7 | 1.1 | 7.2 |
| Spaghetti | 329.0 | 8.7 | 66.3 | 1.3 | 2.8 | 0.4 | 0.9 | 1.4 | 9.4 |
| Penne | 338.4 | 6.9 | 72.4 | 4.3 | 1.9 | 0.3 | 0.5 | 0.9 | 4.9 |
| Lasagne sheets | 373.0 | 7.0 | 76.3 | 0.8 | 4.0 | 1.1 | 1.6 | 0.8 | 2.4 |
| Vermicelli | 371.2 | 12.5 | 71.6 | 2.0 | 3.4 | 0.4 | 1.0 | 1.7 | 13.5 |
| Tagliatelli | 370.9 | 12.1 | 72.0 | 1.9 | 3.3 | 0.4 | 1.0 | 1.7 | 13.1 |
| Cous Cous | 345.0 | 8.8 | 73.8 | 1.5 | 1.1 | 0.1 | 0.4 | 0.4 | 5.0 |
| **Cereals** | | | | | | | | | |
| Granola (chocolate) | 392.3 | 5.5 | 72.6 | 34.0 | 8.5 | 4.7 | 3.1 | 0.6 | 4.6 |
| Granola (nuts) | 478.0 | 7.1 | 64.9 | 16.7 | 21.0 | 7.7 | 10.3 | 2.5 | 4.8 |
| Cornflakes | 322.4 | 8.5 | 62.9 | 1.4 | 3.7 | 0.6 | 1.1 | 1.6 | 7.6 |
| **Cookie and Cakes** | | | | | | | | | |
| Shortbread | 385.3 | 3.3 | 73.6 | 13.8 | 8.3 | 2.7 | 3.9 | 1.5 | 1.2 |
| Neapolitan wafers (original) | 236.0 | 2.5 | 22.9 | 18.1 | 15.0 | 8.3 | 5.2 | 1.1 | 3.3 |
| Cookie (chocolate) | 479.2 | 2.0 | 64.3 | 5.7 | 23.8 | 11.9 | 8.5 | 2.5 | 2.3 |
| Mignon wafers (hazelnut) | 507.9 | 5.0 | 54.0 | 41.7 | 30.4 | 13.9 | 11.9 | 3.7 | 5.6 |
| Marble cake | 403.7 | 5.4 | 48.1 | 20.7 | 22.6 | 3.8 | 10.2 | 7.4 | 0.8 |
| Ladyfinger | 356.9 | 5.7 | 74.6 | 33.1 | 3.5 | 1.0 | 1.4 | 0.6 | 2.5 |
| Cookie (whole-grain) | 471.1 | 4.6 | 71.8 | 21.1 | 18.2 | 7.8 | 6.8 | 2.8 | 3.6 |
| Granola bar | 400.8 | 7.2 | 59.2 | 25.0 | 14.8 | 7.0 | 5.3 | 1.7 | 12.9 |
| Cookie (orange)[a] | 433.0 | 6.2 | 60.2 | 49.7 | 18.3 | 10.5 | 5.5 | 1.4 | 2.6 |
| Apple strudel | 270.9 | 4.2 | 43.2 | 18.1 | 8.7 | 3.4 | 2.9 | 2.2 | 1.6 |
| Muffin | 371.6 | 5.2 | 55.8 | 30.0 | 14.6 | 4.1 | 3.4 | 2.2 | 3.5 |

Table 1 (*continued*)

| | Energy (kcal) | Protein (g) | Carbohydrates (g) | Sugar (g) | Total fat (g) | Saturated fatty acids (g) | MUFA (mg) | PUFA (mg) | Fiber (g) |
|---|---|---|---|---|---|---|---|---|---|
| **Snacks** | | | | | | | | | |
| Cracker | 340.8 | 10.1 | 43.5 | 0.0 | 13.8 | 9.3 | 3.3 | 0.4 | 0.3 |
| Brezels | 449.7 | 0.5 | 65.5 | 0.8 | 20.6 | 9.8 | 7.5 | 2.4 | 0.8 |
| Grissini | 392.5 | 2.0 | 67.4 | 2.6 | 12.6 | 3.9 | 6.1 | 2.4 | 0.7 |
| Saltsticks | 480.9 | 0.6 | 72.0 | 1.5 | 21.1 | 10.3 | 7.8 | 2.2 | 0.8 |
| Wafers (plain) | 329.7 | 8.6 | 63.8 | 1.4 | 4.1 | 0.6 | 1.2 | 1.8 | 7.7 |
| **Convenience foods** | | | | | | | | | |
| Pizza (salami) | 235.1 | 8.0 | 24.8 | 2.8 | 11.5 | 4.8 | 4.1 | 1.9 | 1.5 |
| Pizza (margherita) | 209.1 | 6.3 | 27.6 | 3.5 | 8.0 | 3.8 | 2.5 | 1.3 | 2.0 |
| Lasagne | 170.4 | 7.4 | 16.4 | 2.2 | 8.3 | 3.8 | 3.2 | 0.8 | 1.1 |
| Chicken nuggets | 251.2 | 15.3 | 22.5 | 0.5 | 11.0 | 1.5 | 4.2 | 4.8 | 3.0 |
| Fish sticks | 216.0 | 9.2 | 28.1 | 4.0 | 7.3 | 0.8 | 1.8 | 3.8 | 1.9 |
| Tortellini (pork) | 285.3 | 7.7 | 48.1 | 1.3 | 6.6 | 2.5 | 2.5 | 1.0 | 1.7 |
| Soup (potato and leek) | 355.9 | 16.2 | 47.7 | 7.9 | 10.4 | 2.2 | 4.3 | 2.6 | 8.3 |
| Soup (mushrooms) | 431.4 | 15.5 | 43.4 | 13.7 | 21.7 | 2.1 | 10.2 | 5.7 | 1.6 |
| Wafer-cone (icecream filling) | 278.0 | 5.2 | 27.8 | 24.2 | 16.2 | 5.9 | 7.0 | 2.9 | 3.5 |
| Pudding (semolina) | 362.5 | 6.2 | 82.0 | 14.8 | 0.6 | 0.1 | 0.2 | 0.2 | 1.8 |
| Baked pastry case | 483.9 | 4.3 | 43.3 | 21.6 | 32.9 | 15.7 | 12.0 | 3.6 | 4.4 |
| Wafer (Oblate) | 329.2 | 1.1 | 79.1 | 0.2 | 0.4 | 0.1 | 0.1 | 0.2 | 1.2 |
| Rice drink (natural) | 56.9 | 1.1 | 11.0 | 0.2 | 0.9 | 0.1 | 0.4 | 0.3 | 0.4 |
| Flaky pastry | 389.5 | 1.4 | 31.6 | 2.05 | 29.0 | 8.1 | 7.3 | 12.3 | 5.3 |
| Frozen cake (almond, chocolate) | 405.2 | 7.0 | 25.1 | 24.4 | 31.1 | 10.9 | 13.1 | 4.3 | 2.0 |

**Notes.**

Data displayed as mean values.
[a] Mean values of two very similar products were pooled.

Across all food categories, energy content, carbohydrate, total fat, saturated fatty acids, fiber and sugar did not differ between GF and products gluten-containing products ($F < 1; p > 0.05$).

Protein content was significantly lower in GF foods ($5.8 \pm 3.7$ g/100 g) than gluten-containing foods ($8.6 \pm 2.9$ g/100 g); $F = 31.9; p < 0.01$ (see Fig. 1). Lower protein content was present in 4 out of 7 food categories (flour/bake mix, bread and bakery products, pasta and cereal-based products and snacks). In flour/bake mix products, the average protein content was $4.6 \pm 3.4$ g/100 g for GF and $9.9 \pm 2$ g/100 g for their gluten-containing counterparts (see Table 3).

**Secondary outcome parameter: micronutrients and product cost**

Overall, sodium content in gluten-containing foods ($448.9 \pm 704.6$ mg/100 g) did not differ compared to GF foods ($373.5 \pm 569.2$ mg/100 g; $F < 1, p > 0.05$). In one category (cereal products), sodium content was higher in GF foods. Across all three analyzed GF cereal products, sodium content was $491.3 \pm 91.6$ mg/100 g while in gluten-containing foods, sodium content was $160.7 \pm 139.3$ mg/100 g ($F = 13.4; p < 0.01$). For bread and bakery products, sodium content was lower in GF products ($388.4 \pm 206.4$ mg/100 g) compared to gluten-containing foods ($581.9 \pm 290.3$ mg/100 g; $F = 4.5; p < 0.05$).

Missbach et al. (2015), *PeerJ*, DOI 10.7717/peerj.1337

**Table 2** **Micronutrient composition of gluten-free products in Austria.**

| | Sodium (mg) | Cholestorol (mg) | Iron (mg) | Calcium (mg) | Potassium (mg) | Zinc (mg) | Phosphor (mg) | Vitamin C (mg) | Vitamin D (µg) | Vitamin E (mg) | Retinol (µg) | ß-Carotin (µg) | Thiamin (mg) | Riboflavin (mg) | Niacin (mg) |
|---|---|---|---|---|---|---|---|---|---|---|---|---|---|---|---|
| **Flour/bake mix** | | | | | | | | | | | | | | | |
| Flour[a] | 3.02 | 0.00 | 1.42 | 32.16 | 147.84 | 0.99 | 78.76 | 0.00 | 0.00 | 0.11 | 0.00 | 0.02 | 0.08 | 0.03 | 0.29 |
| Bake mix white (cake) | 39.67 | 0.00 | 1.17 | 54.73 | 240.21 | 1.08 | 204.11 | 0.12 | 0.00 | 0.04 | 0.00 | 0.00 | 0.03 | 0.02 | 0.29 |
| Bake mix brown (cake) | 41.71 | 9.47 | 1.23 | 17.52 | 252.71 | 0.79 | 118.76 | 0.32 | 0.00 | 0.06 | 0.00 | 0.00 | 0.02 | 0.02 | 0.29 |
| Bake mix (Pizza)[a] | 783.75 | 0.00 | 1.49 | 89.31 | 347.84 | 1.64 | 720.85 | 0.39 | 0.00 | 0.08 | 0.00 | 0.00 | 0.06 | 0.05 | 1.14 |
| Breadcrumbs[a] | 196.33 | 0.04 | 2.25 | 31.67 | 182.99 | 1.88 | 183.23 | 0.00 | 0.00 | 1.02 | 0.00 | 4.51 | 0.25 | 0.04 | 0.31 |
| **Bread/bakery products** | | | | | | | | | | | | | | | |
| Rustic bread | 120.53 | 0.00 | 0.38 | 10.58 | 75.45 | 0.48 | 35.90 | 0.00 | 0.00 | 1.01 | 0.00 | 0.00 | 0.06 | 0.02 | 0.48 |
| Whole-grain bread | 685.79 | 0.00 | 2.41 | 96.43 | 304.10 | 1.70 | 208.75 | 0.01 | 0.00 | 0.23 | 0.00 | 0.01 | 0.21 | 0.10 | 1.33 |
| Toast[a] | 394.29 | 0.02 | 1.58 | 34.01 | 273.02 | 1.24 | 135.35 | 0.03 | 0.00 | 2.05 | 0.00 | 0.37 | 0.17 | 0.11 | 1.92 |
| Bun | 402.11 | 4.02 | 0.60 | 17.67 | 52.68 | 0.39 | 36.52 | 0.00 | 0.00 | 0.48 | 0.00 | 1.28 | 0.03 | 0.06 | 0.48 |
| Ciabatta | 355.74 | 0.00 | 1.03 | 17.15 | 117.38 | 0.86 | 91.38 | 1.20 | 0.00 | 1.05 | 0.00 | 0.08 | 0.14 | 0.09 | 0.97 |
| Raisin bread | 299.48 | 0.37 | 1.01 | 59.31 | 157.72 | 0.73 | 80.86 | 0.43 | 0.00 | 0.08 | 0.02 | 0.04 | 0.07 | 0.09 | 0.51 |
| Scone | 314.90 | 32.64 | 0.82 | 30.76 | 91.38 | 0.50 | 73.71 | 0.01 | 0.00 | 0.19 | 0.06 | 0.05 | 0.07 | 0.14 | 1.09 |
| Baguette | 336.38 | 0.00 | 0.66 | 13.96 | 120.93 | 0.41 | 48.25 | 0.84 | 0.00 | 1.41 | 0.00 | 0.01 | 0.07 | 0.06 | 0.62 |
| Lye Pretzel | 790.76 | 7.06 | 0.40 | 124.30 | 137.68 | 0.57 | 78.43 | 13.10 | 0.00 | 0.49 | 0.02 | 1.29 | 0.04 | 0.15 | 0.15 |
| Rusk | 5.52 | 0.01 | 1.06 | 21.34 | 11.95 | 0.40 | 20.24 | 0.00 | 0.00 | 0.08 | 0.00 | 1.00 | 0.00 | 0.01 | 0.09 |
| Crispbread | 547.20 | 0.00 | 0.67 | 11.61 | 99.13 | 0.67 | 109.04 | 0.00 | 0.00 | 0.24 | 0.00 | 0.04 | 0.11 | 0.04 | 1.38 |
| Wraps | 402.35 | 0.63 | 0.21 | 18.09 | 83.57 | 0.19 | 39.96 | 0.25 | 0.23 | 0.07 | 0.05 | 0.06 | 0.03 | 0.01 | 0.58 |
| **Pasta and cereal-based products** | | | | | | | | | | | | | | | |
| Fusilli | 1.88 | 0.00 | 1.80 | 14.67 | 114.68 | 1.87 | 205.75 | 0.00 | 0.00 | 0.81 | 0.00 | 0.21 | 0.33 | 0.10 | 1.76 |
| Spaghetti | 1.00 | 0.00 | 2.40 | 18.00 | 120.00 | 2.50 | 256.00 | 0.00 | 0.00 | 1.11 | 0.00 | 0.30 | 0.44 | 0.13 | 1.93 |
| Penne | 1.90 | 0.00 | 3.15 | 22.73 | 198.37 | 1.43 | 211.40 | 0.00 | 0.00 | 0.48 | 0.00 | 0.12 | 0.28 | 0.12 | 1.61 |
| Lasagne sheets | 55.98 | 146.52 | 1.37 | 30.53 | 91.97 | 1.32 | 144.47 | 0.00 | 0.00 | 0.89 | 0.10 | 0.04 | 0.10 | 0.18 | 0.52 |
| Vermicelli | 6.71 | 0.00 | 3.81 | 45.15 | 297.12 | 3.05 | 327.38 | 0.93 | 0.00 | 1.68 | 0.00 | 0.31 | 0.51 | 0.16 | 2.14 |
| Tagliatelli | 5.94 | 0.00 | 3.65 | 41.70 | 274.68 | 3.02 | 321.78 | 0.80 | 0.00 | 1.62 | 0.00 | 0.32 | 0.51 | 0.16 | 2.14 |
| Cous Cous | 1.00 | 0.00 | 1.00 | 4.00 | 80.00 | 0.41 | 73.00 | 0.00 | 0.00 | 0.52 | 0.00 | 0.26 | 0.13 | 0.04 | 1.20 |
| **Cereals** | | | | | | | | | | | | | | | |
| Granola (chocolate) | 504.01 | 15.77 | 1.51 | 17.98 | 265.98 | 0.52 | 73.86 | 0.13 | 0.00 | 0.25 | 0.00 | 0.51 | 0.05 | 0.06 | 0.96 |
| Granola (nuts) | 393.96 | 0.16 | 1.72 | 44.88 | 314.76 | 1.60 | 174.62 | 0.11 | 0.00 | 3.54 | 0.00 | 1.78 | 0.20 | 0.09 | 1.40 |
| Cornflakes | 575.88 | 0.00 | 1.52 | 8.61 | 265.44 | 1.45 | 208.76 | 0.00 | 0.00 | 1.47 | 0.00 | 0.90 | 0.35 | 0.20 | 1.47 |
| **Cookie and cakes** | | | | | | | | | | | | | | | |
| Shortbread | 408.14 | 17.31 | 0.66 | 33.19 | 79.97 | 0.59 | 72.93 | 0.22 | 0.00 | 0.84 | 0.02 | 9.02 | 0.05 | 0.10 | 0.73 |
| Neapolitan wafers (original) | 16.20 | 21.70 | 0.71 | 8.86 | 259.18 | 0.59 | 61.06 | 0.01 | 0.00 | 0.12 | 0.00 | 0.01 | 0.03 | 0.03 | 0.37 |
| Cookie (chocolate) | 199.30 | 3.37 | 0.82 | 13.20 | 91.10 | 0.73 | 57.96 | 0.00 | 0.00 | 1.66 | 0.00 | 4.50 | 0.05 | 0.02 | 0.26 |
| Mignon wafers (hazelnut) | 417.91 | 28.84 | 2.70 | 28.12 | 476.92 | 1.16 | 124.82 | 0.02 | 0.00 | 1.31 | 0.00 | 0.03 | 0.09 | 0.06 | 0.68 |
| Marble cake | 54.88 | 138.60 | 1.32 | 22.11 | 103.26 | 0.76 | 98.91 | 0.00 | 0.00 | 12.54 | 0.10 | 0.02 | 0.05 | 0.15 | 0.24 |
| Ladyfinger | 98.46 | 141.37 | 1.00 | 37.26 | 79.88 | 0.80 | 100.96 | 0.00 | 0.00 | 0.72 | 0.10 | 0.00 | 0.05 | 0.15 | 0.28 |
| Cookie (whole-grain) | 295.19 | 5.16 | 1.17 | 15.24 | 68.83 | 0.78 | 89.30 | 0.13 | 0.00 | 0.37 | 0.00 | 0.12 | 0.14 | 0.05 | 0.87 |
| Granola bar | 237.91 | 4.24 | 2.14 | 96.16 | 292.51 | 1.37 | 170.55 | 0.45 | 0.00 | 1.00 | 0.01 | 0.16 | 0.19 | 0.19 | 1.23 |
| Cookie (orange)[a] | 190.66 | 23.28 | 1.74 | 103.46 | 389.10 | 0.83 | 120.50 | 4.78 | 0.00 | 0.39 | 0.06 | 0.27 | 0.08 | 0.13 | 0.45 |
| Apple strudel | 102.98 | 1.83 | 1.13 | 177.03 | 240.82 | 0.68 | 128.36 | 4.75 | 0.00 | 0.16 | 0.05 | 0.08 | 0.07 | 0.27 | 0.25 |
| Muffin | 247.84 | 79.20 | 2.66 | 29.50 | 219.98 | 0.74 | 102.31 | 0.00 | 0.00 | 3.54 | 0.06 | 0.01 | 0.06 | 0.11 | 0.30 |

Missbach et al. (2015), *PeerJ*, DOI 10.7717/peerj.1337

Table 2 (*continued*)

| | Sodium (mg) | Cholestorol (mg) | Iron (mg) | Calcium (mg) | Potassium (mg) | Zinc (mg) | Phosphor (mg) | Vitamin C (mg) | Vitamin D (µg) | Vitamin E (mg) | Retinol (µg) | ß-Carotin (µg) | Thiamin (mg) | Riboflavin (mg) | Niacin (mg) |
|---|---|---|---|---|---|---|---|---|---|---|---|---|---|---|---|
| **Snacks** | | | | | | | | | | | | | | | |
| Cracker | 2416.36 | 47.71 | 1.40 | 434.99 | 83.08 | 2.05 | 305.78 | 0.43 | 0.00 | 0.41 | 0.13 | 0.06 | 0.03 | 0.12 | 0.48 |
| Brezels | 554.28 | 0.20 | 0.67 | 11.72 | 17.23 | 0.45 | 26.93 | 0.00 | 0.00 | 1.48 | 0.00 | 4.26 | 0.01 | 0.03 | 0.21 |
| Grissini | 226.87 | 20.01 | 0.63 | 52.59 | 64.73 | 0.62 | 62.82 | 0.06 | 0.00 | 1.31 | 0.03 | 15.02 | 0.02 | 0.09 | 0.07 |
| Saltsticks | 1007.36 | 0.21 | 0.93 | 15.82 | 25.22 | 0.47 | 30.69 | 0.00 | 0.00 | 1.56 | 0.00 | 4.47 | 0.02 | 0.04 | 0.34 |
| Wafers (plain) | 119.98 | 0.01 | 1.50 | 8.15 | 268.55 | 1.47 | 211.73 | 0.02 | 0.00 | 1.61 | 0.00 | 0.92 | 0.36 | 0.20 | 1.49 |
| **Convenience foods** | | | | | | | | | | | | | | | |
| Pizza (salami) | 614.15 | 22.66 | 0.97 | 103.75 | 275.57 | 1.13 | 115.28 | 7.43 | 0.00 | 3.23 | 0.04 | 0.24 | 0.11 | 0.11 | 1.68 |
| Pizza (margherita) | 437.51 | 14.95 | 0.68 | 110.73 | 266.87 | 0.85 | 106.10 | 6.97 | 0.00 | 2.97 | 0.05 | 0.26 | 0.05 | 0.10 | 1.02 |
| Lasagne | 450.21 | 38.88 | 0.79 | 56.08 | 211.47 | 1.38 | 99.60 | 5.51 | 0.02 | 1.14 | 0.04 | 0.24 | 0.11 | 0.13 | 1.51 |
| Nuggets | 396.19 | 35.00 | 1.34 | 16.80 | 158.47 | 1.37 | 179.53 | 0.05 | 0.00 | 5.46 | 0.00 | 0.16 | 0.14 | 0.09 | 4.71 |
| Fish sticks | 378.34 | 23.54 | 1.19 | 24.87 | 128.33 | 0.49 | 142.50 | 0.24 | 0.00 | 4.04 | 0.00 | 0.11 | 0.12 | 0.22 | 1.62 |
| Tortellini (pork) | 628.05 | 72.33 | 1.49 | 51.06 | 118.68 | 1.64 | 122.19 | 1.06 | 0.01 | 0.75 | 0.04 | 0.04 | 0.24 | 0.12 | 1.38 |
| Soup (potato and leek) | 3801.76 | 1.03 | 2.53 | 80.57 | 1014.25 | 0.88 | 260.47 | 11.70 | 0.00 | 1.61 | 0.00 | 0.42 | 0.17 | 0.28 | 4.63 |
| Soup (mushrooms) | 472.57 | 0.43 | 0.89 | 17.52 | 104.01 | 0.22 | 34.90 | 0.95 | 0.00 | 3.82 | 0.00 | 0.68 | 0.04 | 0.07 | 0.91 |
| Wafer-cone (icecream filling) | 101.42 | 22.61 | 1.39 | 97.43 | 358.09 | 0.78 | 127.37 | 0.81 | 0.12 | 3.66 | 0.02 | 0.04 | 0.04 | 0.16 | 0.37 |
| Pudding (semolina) | 117.48 | 0.00 | 0.54 | 9.64 | 91.27 | 0.50 | 79.69 | 0.10 | 0.00 | 0.09 | 0.00 | 0.00 | 0.06 | 0.02 | 1.18 |
| Baked pastry case | 108.30 | 0.32 | 1.61 | 30.08 | 138.24 | 1.30 | 202.23 | 0.70 | 0.00 | 2.87 | 0.00 | 6.82 | 0.18 | 0.09 | 1.91 |
| Wafer (Oblate) | 6.98 | 0.00 | 1.80 | 31.88 | 26.15 | 0.43 | 35.54 | 0.00 | 0.00 | 0.12 | 0.00 | 0.03 | 0.05 | 0.02 | 0.33 |
| Rice drink (natural) | 52.58 | 0.00 | 0.23 | 12.47 | 17.01 | 0.20 | 15.55 | 0.00 | 0.00 | 0.51 | 0.00 | 0.00 | 0.01 | 0.00 | 0.19 |
| Flaky pastry | 412.00 | 2.35 | 0.22 | 49.65 | 47.07 | 0.20 | 24.56 | 0.13 | 0.03 | 2.10 | 0.00 | 0.17 | 0.24 | 0.01 | 0.22 |
| Frozen cake (almond, chocolate) | 120.14 | 170.20 | 1.69 | 39.44 | 219.39 | 0.90 | 126.96 | 0.21 | 0.13 | 4.26 | 0.15 | 2.19 | 0.06 | 0.16 | 0.38 |

**Notes.**

Data displayed as mean values.

[a] Mean values of two very similar products were pooled.

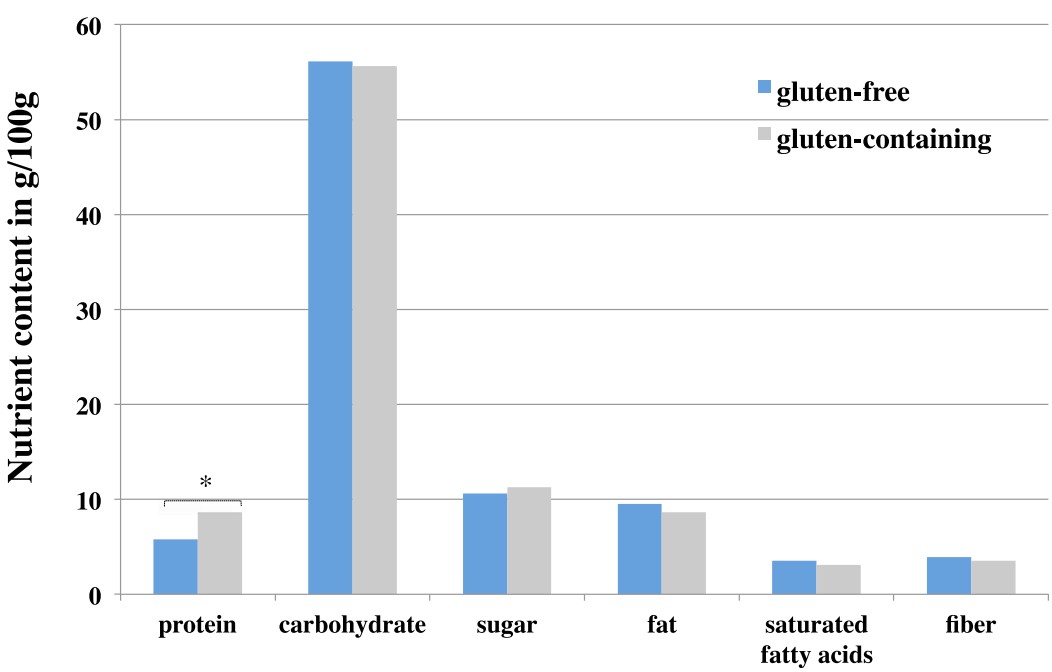

**Figure 1  Nutrient content in g/100 g between gluten-free and gluten-containing foods across seven different food categories.** Notes. Data displayed as mean values. * Significant differences ($p < 0.05$) between gluten-free and gluten containing foods.

Across all other categories, sodium content did not differ significantly. 27% of all products showed high sodium content (defined as >500 mg/100 g, *Commission of the European Communities (2006)*),  this did not differ between GF and gluten-containing foods ($\chi^2$ [1] $= 1.94; p > 0.05$). In contrast, 65% of GF and 61% of gluten-containing foods showed low sodium content (defined as <120 mg/100 g, *Commission of the European Communities (2006)*).

Potassium content was significantly lower in GF products (190.4 $\pm$ 160 mg/100 g) than in products containing gluten (247.5 $\pm$ 130 mg/100 g; $F = 6.9; p < 0.05$). This difference was present in 2 out of 7 food categories (pasta and cereal-based products; snacks), while all other categories did no show significant differences. Moreover, zinc content was significantly lower in GF pasta products (1.9 $\pm$ 0.9 mg/100 g vs. 4.6 $\pm$ 0.4 mg/100 g; $F = 82.1; p < 0.01$). GF Pasta products showed higher fiber content (7.9 $\pm$ 4.2 g/100 g) when compared to gluten-containing products (3.7 $\pm$ 0.7 g/100 g; $F = 13.6$; $p < 0.01$). Across all GF products, 19% can be classified as source high in fiber (defined as >6 g/100 g, *Commission of the European Communities (2006)*).

Substantial cost disparities were present between GF products and gluten-containing products. On average, the cost for GF products ranged from 2.95€ (white flour) to 80.80€ per kg (Wafer, Oblaten) and was significantly higher in GF products (11.58 $\pm$ 11.43€) compared to gluten-containing products (6.62 $\pm$ 5.36€; $F = 53.1; p < 0.01$) across all product categories. Within bread and bakery products, GF food were +267% more

**Table 3** Comparison between micro-and macronutrient composition of gluten-free and matched gluten-containing foods among categories.

| | | Flour/bake mix | Bread/bakery products | Pasta and cereal-based products | Cereals | Cookie and cakes | Snacks | Convenience foods |
|---|---|---|---|---|---|---|---|---|
| Energy (kcal) | Gluten free | 346.4 ± 35.5 | 270.5 ± 46.7 | 351.9 ± 17.7 | 397.5 ± 63.6 | 395.8 ± 84.4 | 398.7 ± 59.2 | 297.3 ± 108.5 |
| | Gluten-containing | 335.0 ± 23.0 | 280.9 ± 49.8 | 346.5 ± 10.9 | 397.4 ± 56.3 | 416.3 ± 78.1 | 371.6 ± 45.3 | 298.2 ± 102.0 |
| | P† | 0.373 | 0.543 | 0.426 | 0.997 | 0.488 | 0.376 | 0.978 |
| Protein (g) | Gluten Free | 4.6 ± 3.2 | 4.1 ± 2.2 | 9.2 ± 2.1 | 7.0 ± 1.2 | 4.8 ± 1.6 | 4.3 ± 4.1 | 7.4 ± 4.8 |
| | Gluten-containing | 10.0 ± 2.0 | 8.3 ± 1.0 | 11.9 ± 0.9 | 9.2 ± 2.4 | 5.8 ± 1.9 | 10.5 ± 1.4 | 8.2 ± 3.9 |
| | P† | <0.01 | <0.01 | <0.05 | 0.245 | 0.105 | <0.01 | 0.605 |
| Carbohydrates (g) | Gluten Free | 74.4 | 52.3 | 71.7 | 66.7 | 57.3 | 62.4 | 37.2 |
| | Gluten-containing | 67.8 | 52.7 | 69.4 | 57.6 | 59.9 | 61.4 | 39.5 |
| | P† | <0.05 | 0.912 | <0.05 | 0.101 | 0.561 | 0.860 | 0.733 |
| Sugar (g) | Gluten Free | 9.5 ± 17.2 | 5.5 ± 5.2 | 1.8 ± 1.1 | 17.4 ± 13.3 | 27.2 ± 14.6 | 1.3 ± 0.8 | 8.2 ± 8.8 |
| | Gluten-containing | 6.0 ± 9.1 | 4.4 ± 5.4 | 0.8 ± 0.2 | 14.5 ± 6.4 | 33.7 ± 12.6 | 1.0 ± 0.2 | 9.8 ± 12.2 |
| | P† | 0.543 | 0.912 | <0.05 | 0.714 | 0.188 | <0.860 | 0.655 |
| Total fat (g) | Gluten Free | 2.9 ± 2.2 | 4.7 ± 2.7 | 2.7 ± 0.9 | 11.1 ± 7.3 | 16.4 ± 8.4 | 14.4 ± 6.2 | 13.1 ± 10.5 |
| | Gluten-containing | 2.2 ± 2.3 | 3.8 ± 2.8 | 1.9 ± 1.0 | 14.3 ± 8.0 | 17.0 ± 8.4 | 9.0 ± 8.0 | 11.6 ± 9.0 |
| | P† | 0.481 | 0.349 | 0.133 | 0.621 | 0.852 | 0.238 | 0.630 |
| Saturated fatty acids (g) | Gluten Free | 0.9 ± 1.1 | 1.4 ± 1.3 | 0.4 ± 0.3 | 4.3 ± 2.9 | 7.1 ± 5.1 | 6.8 ± 3.8 | 4.1 ± 4.3 |
| | Gluten-containing | 0.7 ± 1.2 | 0.9 ± 1.0 | 0.4 ± 0.3 | 4.4 ± 3.4 | 6.8 ± 4.5 | 4.2 ± 5.3 | 4.0 ± 4.1 |
| | P† | 0.716 | 0.183 | 0.818 | 0.986 | 0.868 | 0.384 | 0.917 |
| MUFA (mg) | Gluten Free | 1.0 ± 1.1 | 1.5 ± 0.8 | 0.9 ± 0.4 | 4.8 ± 4.0 | 5.9 ± 3.2 | 5.2 ± 2.6 | 4.8 ± 4.1 |
| | Gluten-containing | 0.5 ± 0.9 | 1.3 ± 1.2 | 0.4 ± 0.4 | 6.4 ± 4.1 | 6.7 ± 3.6 | 2.7 ± 2.4 | 4.5 ± 3.7 |
| | P† | 0.236 | 0.614 | <0.05 | 0.627 | 0.509 | 0.117 | 0.783 |
| PUFA (mg) | Gluten Free | 0.8 ± 0.6 | 1.4 ± 1.1 | 1.1 ± 0.4 | 1.5 ± 0.8 | 2.4 ± 1.7 | 1.8 ± 0.7 | 3.0 ± 3.0 |
| | Gluten-containing | 0.6 ± 0.4 | 1.0 ± 0.8 | 0.8 ± 0.3 | 3.2 ± 0.9 | 2.6 ± 1.8 | 1.4 ± 1.0 | 1.9 ± 1.4 |
| | P† | 0.220 | 0.195 | 0.100 | 0.055 | 0.723 | 0.412 | 0.101 |
| Fiber (g) | Gluten Free | 4.1 ± 2.9 | 3.9 ± 2.7 | 7.9 ± 4.0 | 5.6 ± 1.4 | 3.5 ± 3.1 | 2.0 ± 2.8 | 2.6 ± 2.0 |
| | Gluten-containing | 4.0 ± 2.3 | 3.3 ± 1.7 | 3.7 ± 0.7 | 7.4 ± 1.9 | 3.5 ± 3.8 | 4.6 ± 3.2 | 2.4 ± 2.0 |
| | P† | 0.944 | 0.429 | <0.05 | 0.247 | 0.979 | 0.188 | 0.669 |
| Sodium (mg) | Gluten Free | 255.9 ± 326.9 | 388.4 ± 198.3 | 10.6 ± 18.6 | 491.3 ± 74.8 | 205.0 ± 122.6 | 856.0 ± 835.0 | 539.8 ± 894.2 |
| | Gluten-containing | 281.6 ± 294.6 | 581.9 ± 284.6 | 15.8 ± 18.0 | 160.7 ± 127.2 | 247.8 ± 394.0 | 832.1 ± 626.7 | 715.8 ± 1186.1 |
| | P† | 0.855 | 0.039 | 0.564 | <0.05 | 0.724 | 0.938 | 0.623 |
| Cholestorol (mg) | Gluten Free | 1.2 ± 3.1 | 3.4 ± 8.7 | 20.9 ± 51.3 | 5.3 ± 7.4 | 40.7 ± 49.0 | 13.6 ± 18.7 | 27.0 ± 43.1 |
| | Gluten-containing | 2.6 ± 7.0 | 5.5 ± 12.1 | 21.5 ± 42.2 | 1.4 ± 2.0 | 32.4 ± 38.4 | 1.5 ± 2.6 | 28.5 ± 41.5 |
| | P† | 0.617 | 0.602 | 0.980 | 0.318 | 0.595 | 0.083 | 0.908 |
| Iron (mg) | Gluten Free | 1.6 ± 0.8 | 1.0 ± 0.7 | 2.5 ± 1.0 | 1.6 ± 0.1 | 1.5 ± 0.7 | 1.0 ± 0.4 | 1.2 ± 0.6 |
| | Gluten-containing | 1.6 ± 1.1 | 1.2 ± 0.6 | 2.5 ± 0.3 | 6.1 ± 7.3 | 1.6 ± 1.3 | 2.2 ± 1.8 | 1.4 ± 1.3 |
| | P† | 0.964 | 0.216 | 0.944 | 0.375 | 0.840 | 0.188 | 0.496 |
| Calcium (mg) | Gluten Free | 47.3 ± 27.2 | 37.6 ± 33.9 | 25.3 ± 13.7 | 23.8 ± 15.4 | 55.6 ± 55.0 | 104.7 ± 165.9 | 48.8 ± 33.1 |
| | Gluten-containing | 24.3 ± 19.1 | 18.9 ± 10.8 | 33.6 ± 7.6 | 75.4 ± 117.0 | 37.2 ± 27.5 | 50.3 ± 45.1 | 46.4 ± 40.0 |

| | | Flour/bake mix | Bread/bakery products | Pasta and cereal-based products | Cereals | Cookie and cakes | Snacks | Convenience foods |
|---|---|---|---|---|---|---|---|---|
| | P† | <0.05 | <0.05 | 0.105 | 0.524 | 0.202 | 0.384 | 0.842 |
| Potassium (mg) | Gluten Free | 231.3 ± 127.5 | 138.3 ± 118.9 | 168.1 ± 82.5 | 282.1 ± 23.1 | 224.2 ± 136.7 | 91.8 ± 91.7 | 211.7 ± 234.1 |
| | Gluten-containing | 222.3 ± 86.8 | 201.3 ± 61.0 | 295.4 ± 121.2 | 378.3 ± 152.7 | 235.1 ± 124.3 | 309.7 ± 179.8 | 241.8 ± 145.4 |
| | P† | 0.848 | <0.05 | <0.05 | 0.370 | 0.818 | 0.352 | 0.607 |
| Zinc (mg) | Gluten Free | 1.4 ± 0.8 | 0.7 ± 0.5 | 1.9 ± 0.9 | 1.2 ± 0.5 | 0.8 ± 0.2 | 1.0 ± 0.6 | 0.8 ± 0.5 |
| | Gluten-containing | 1.2 ± 1.1 | 0.8 ± 0.4 | 4.6 ± 0.4 | 2.6 ± 1.0 | 0.8 ± 0.6 | 1.6 ± 1.1 | 1.0 ± 0.8 |
| | P† | 0.734 | 0.766 | <0.01 | 0.078 | 0.930 | 0.370 | 0.346 |
| Phosphor (mg) | Gluten Free | 286.1 ± 267.0 | 84.1 ± 61.3 | 220.0 ± 85.2 | 152.4 ± 57.3 | 104.0 ± 33.5 | 127.6 ± 111.8 | 111.5 ± 66.4 |
| | Gluten-containing | 194.7 ± 177.1 | 104.9 ± 49.2 | 301.3 ± 178.4 | 321.0 ± 222.1 | 109.2 ± 73.7 | 207.2 ± 152.4 | 130.7 ± 84.6 |
| | P† | 0.350 | 0.247 | 0.290 | 0.292 | 0.823 | 0.352 | 0.456 |
| Vitamin C (mg) | Gluten Free | 0.2 ± 0.2 | 1.2 ± 3.4 | 0.2 ± 0.4 | 0.1 ± 0.1 | 1.3 ± 2.4 | 0.1 ± 0.2 | 2.4 ± 3.5 |
| | Gluten-containing | 0.0 ± 0.0 | 1.6 ± 7.6 | 1.0 ± 3.5 | 22.4 ± 49.9 | 2.0 ± 3.4 | 0.0 ± 0.1 | 16.7 ± 56.1 |
| | P† | <0.05 | 0.885 | 0.612 | 0.516 | 0.512 | 0.404 | 0.341 |
| Vitamin E (mg) | Gluten Free | 0.3 ± 0.4 | 0.7 ± 0.8 | 1.0 ± 0.4 | 1.8 ± 1.4 | 1.9 ± 3.3 | 1.3 ± 0.4 | 2.4 ± 1.6 |
| | Gluten-containing | 0.3 ± 0.4 | 0.8 ± 1.0 | 1.0 ± 0.2 | 2.1 ± 1.2 | 2.3 ± 3.2 | 0.9 ± 0.5 | 1.7 ± 1.1 |
| | P† | 0.952 | 0.841 | 0.972 | 0.728 | 0.724 | 0.178 | 0.065 |
| ß-Carotin (mg) | Gluten Free | 1.1 ± 3.0 | 0.4 ± 0.5 | 0.2 ± 0.1 | 1.1 ± 0.5 | 1.2 ± 2.7 | 4.9 ± 5.3 | 0.8 ± 1.7 |
| | Gluten-containing | 0.0 ± 0.0 | 0.2 ± 0.5 | 0.1 ± 0.3 | 1.2 ± 1.2 | 1.1 ± 1.6 | 0.4 ± 0.7 | 0.9 ± 1.5 |
| | P† | 0.160 | 0.347 | 0.320 | 0.838 | 0.887 | <0.05 | 0.786 |
| Niacin (mg) | Gluten Free | 0.6 ± 0.4 | 0.9 ± 0.7 | 1.6 ± 0.5 | 1.3 ± 0.2 | 0.5 ± 0.3 | 0.5 ± 0.5 | 1.5 ± 1.4 |
| | Gluten-containing | 1.5 ± 1.6 | 1.6 ± 0.8 | 3.6 ± 2.7 | 11.0 ± 22.0 | 0.7 ± 0.8 | 3.3 ± 3.1 | 1.8 ± 1.6 |
| | P† | 0.128 | <0.05 | 0.077 | 0.523 | 0.552 | 0.084 | 0.488 |

**Notes.**

Data displayed as mean values per unit/100 g ± standard deviation.

$P$† Differences in mean nutrient content between GF and non-GF products were assessed by unpaired $t$-test.

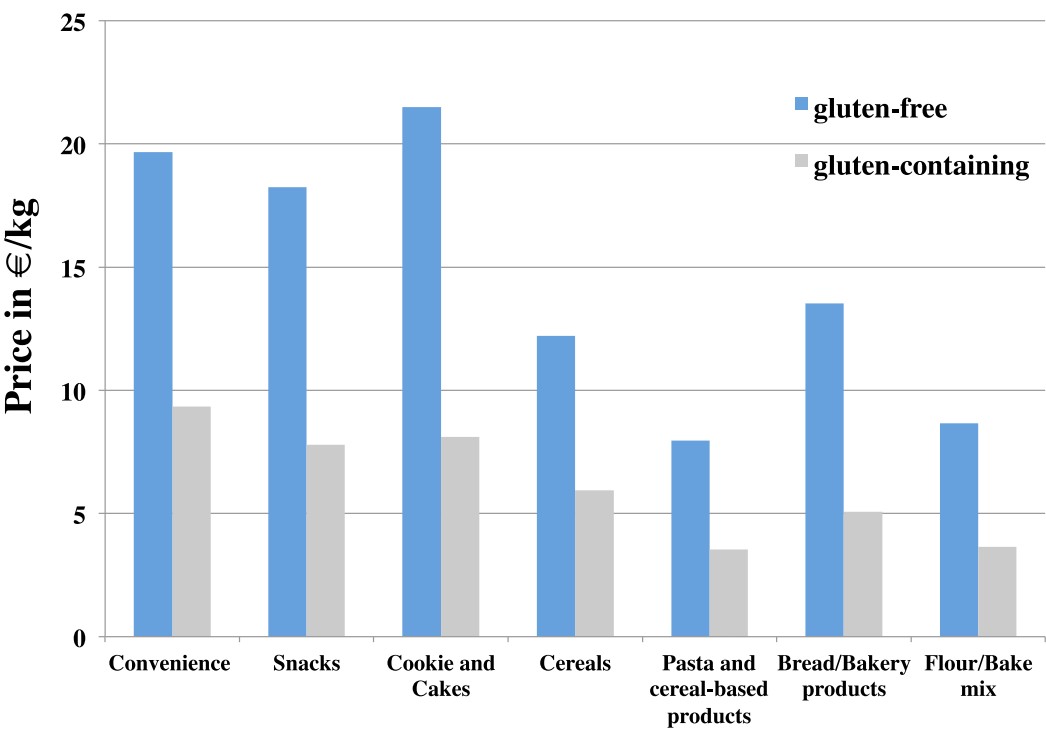

**Figure 2 Cost in €/kg between gluten-free and gluten-containing foods across seven different food categories.** Notes. Data displayed as mean values. Within all food categories differences in mean cost between GF and and gluten-containing foods were significant ($p < 0.05$).

expensive than similar gluten-containing products. The lowest cost disparity (+205% higher cost for GF products) was observed for cereal products (Fig. 2).

## DISCUSSION

The present study is the first attempt to present a large dataset comparing GF foods and gluten-containing products available in one German-speaking country (Austria). The data showed that there is great variability between GF foods and gluten-containing products for specific nutrients.

One key finding of this study is that protein content was significantly lower in GF foods across all staple foods. In flour/bake mix products, the average protein content was >2 fold lower compared to their gluten-containing counterparts. This finding is in line with previous findings (*Wu et al., 2015*), except that we did not observe significant differences in total fat, saturated fat, PUFA and MUFA in our products (*Kulai & Rashid, 2014*; *Matos Segura & Rosell, 2011*; *Miranda et al., 2014*). Only in pasta and cereal-based products was MUFA content significantly higher in GF foods. The low amount of proteins in GF foods can be explained by their formulation. In GF formulations, carbohydrate-rich but protein-poor ingredients are used, such as white rice flour, tapioca or potato starch (*Mezaize et al., 2009*). This can lead to lower protein content in GF foods, which may be a reasonable explanation for the observed differences in our data.

Reports about protein intake and its clinical relevance for celiac patients is conflicting. In a prospective study comparing dietary intake from 88 celiac patients (7-day dietary

record) with data from non-celiac individuals from the German National Diet and Nutrition Survey (NVS II), no differences in protein intake for males or females were observed (*Martin et al., 2013*). On the other hand, *Miranda et al. (2014)* analyzed 58 adults with CD and showed that protein intake was lower in women who were on a GF diet compared to a diet containing gluten. In this study, the protein content of breads was almost one third lower than their equivalent foods with gluten. In our dataset, GF breads contained half of the proteins compared to regular breads with gluten. Addtionally, in a cross-sectional study, *Van Hees et al. (2015)* compared dietary intake of amino acids in 77 CD patients. Compared to 33 healthy controls, celiac patients with good adherence to a GF diet showed significantly lower amino acid concentrations in blood (tyrosine, phenylalanine and tryptophan). The authors argue that both, a reduced intake of vegetable protein and malabsorption as a results of CD may be responsible for this result. The findings of our study suggest that reduced protein content in GF products may facilitate problematic protein intake in CD patients and should be considered in dietary counseling.

In 65% of all analyzed GF foods, low sodium content (defined as <120 mg/100 g, *Commission of the European Communities (2006)*) was observed. Interestingly, in bread and bakery products, sodium content was lower compared to gluten-containing foods. The lower amount of sodium in GF bread may be accounted for the joint initiative "Weniger Salz is g'sünder" with the aim to reduce salt in bread and bakery products by 15% by 2015 initiated by the *Austrian Ministry of Health (2011)* and the Industrial Bakers of Austria (*Lloyd-Williams et al., 2014*). Foods from the datasets used in this study contained nutrient information that were assessed prior to this initiative (started 2011), which may be a possible explanation for this discrepancy.

## LIMITATIONS

Some limitations of the present study should be taken into account. First, we did not analyze the nutritional composition of GF foods through direct chemical analysis, but only estimated the data from nutrient content by deriving data from two nutrient databases. Direct chemical analysis is the gold standard to estimate the nutrient composition of food. Nevertheless, previous studies have shown that estimating the nutrient composition of GF food via indirect analysis is a valid method, likewise (*Mazzeo et al., 2015*; *Miranda et al., 2014*; *Wu et al., 2015*). Additionally, nutrient data shown on food labels provided by the food industry are commonly based on estimation of nutrient content of the ingredients rather than direct chemical analysis of the food products (*Pennington, 2008*).

A second limitiation of the study is the small sample size of the analyzed products. Due to the rigorous exclusion steps applied in this study, we only analyzed 63 from originally 162 identified GF foods. We analyzed foods that are originally based on cereal formulations. In some categories, low numbers of GF foods were included (e.g., category cereals only three items; category flour/bake mix only five items). Hence, post-hoc power analysis revealed that in the case of e.g protein content in GF and gluten-containing food groups in flour/bake mix products, statistical power ($1-\beta$) was still high at 95.7%. Nevertheless, this is only the first step to build a database for GF products in Austria,

and we will be extending the database for future investigations. Therefore, the provided number of foods is a solid starting point for further analysis.

Finally, it should be noted that we only included data from products sold in one German-speaking country (Austria), while a majority of GF products are well distributed across European countries, translating our findings to other countries should be interpreted conservatively. Nevertheless, this study improves our knowledge about the nutritional quality of GF foods and secondly, the applied methodological strategy holds a great potential to consolidate data from other countries to form a transnational database on GF products.

## Implications of the present research

To put our findings into perspective about the ongoing discussion if choosing GF foods holds potential nutritional advantages or disadvantages for consumers, it is important to note that GF products are very popular among consumers without CD. In fact, GF foods are increasingly purchased by individuals without CD (*Silvester et al., 2015*). A report by *Dunn, House & Shelnutt (2014)* showed that only 57% consume GF foods for medical reasons, while for almost half of the consumers other factors e.g., lifestyle and positive health association are important for purchasing GF foods. This trend is reflected in worldwide sales numbers as well. Between 2004 and 2011 the market for GF products grew at an annual growth rate of 28% (*Sapone et al., 2012*). The global GF product market is projected to reach a value of $6,206 million, growing at a compounded annual growth rate of 10.2% by 2018 (*Marketsandmarkets.com, 2013*). This implies that the GF product market represents a very prosperous market in food and beverages. In addition, there is an ongoing discussion about the prevalence of nonceliac gluten sensitivity (NCGS) in the general population. For individuals suffering from NCGS, adhering to a GF diet could also be beneficial in the remission of their symptoms (*El-Chammas & Danner, 2011*). Nevertheless, the majority of clinical evidence for NCGS remain inconsistent and rather controversial (*Biesiekierski, Muir & Gibson, 2013*). From a public health perspective, there is no need to adhere to a GF diet for consumers without diagnosed CD (*Catassi et al., 2013*; *De Giorgio, Volta & Gibson, 2015*; *Molina-Infante et al., 2015*).

Still, the question why GF products are perceived as healthier for consumers without CD is of relevance (*Dunn, House & Shelnutt, 2014*). On a behavioral level, the increased perceived healthfulness may be explained by the 'health halo' effect, which states that products that are labelled as 'healthier' (e.g. low-fat label) can mislead consumers about other important nutritional elements, e.g., energy content and portion sizes (*Faulkner et al., 2014*). The 'health halo' effect can also lead to some undesired behavioral effects such as increased consumption and poor caloric estimates (*Ebneter, Latner & Nigg, 2013*).

Marketers tap into the perceived healthfullness which reflects, besides the increased production cost of GF products, in the overall higher cost of GF products. In our dataset, the cost for all analyzed GF products was 205–267% higher than for conventional foods. This finding is in line with previous findings (*Kulai & Rashid, 2014*; *Lee et al., 2007*; *Singh &*

*Whelan, 2011*; *Stevens & Rashid, 2008*). In fact, *Singh & Whelan (2011)* report even higher cost disparities for GF foods ranging from 70–510%.

## CONCLUSIONS

In conclusion, this study presents the first findings for a thorough analysis of GF products in a German-speaking country. There are some marked differences between GF and gluten-containing foods. Based on the nutrient composition of GF foods, our results indicate that GF foods are not aligned with particular health benefits, but rather show critical nutrients which should be considered in future formulations. The findings of our study indicate that re-thinking the health aspects ascribed to GF products, at least based on nutrient content of GF foods, should be considered and publicly communicated. Especially in the face of a growing market share, common health misconceptions should be kept in mind when discussing GF products.

### Funding

The authors received no funding for this work.

### Competing Interests

The authors declare there are no competing interests.

### Author Contributions

- Benjamin Missbach conceived and designed the experiments, performed the experiments, analyzed the data, contributed reagents/materials/analysis tools, wrote the paper, prepared figures and/or tables, reviewed drafts of the paper.
- Lukas Schwingshackl and Jürgen König conceived and designed the experiments, contributed reagents/materials/analysis tools, reviewed drafts of the paper.
- Alina Billmann, Aleksandra Mystek and Melanie Hickelsberger performed the experiments, reviewed drafts of the paper.
- Gregor Bauer conceived and designed the experiments, reviewed drafts of the paper.

### Supplemental Information

Supplemental information for this article can be found online at http://dx.doi.org/10.7717/peerj.1337#supplemental-information.

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
