# Peer review of "Gluten-free food database: the nutritional quality and cost of packaged gluten-free foods"

_PeerJ, doi:10.7717/peerj.1337_

## Round 0.1 · original submission · Major Revisions

· Academic Editor

Major Revisions

Dear Dr Missbach,

Expert reviewers in the field have evaluated your manuscript, and based on their comments it is not suitable for publication in PeerJ as it currently stands.
We invite you to submit a revised version of the manuscript that addressed all the points raised by the reviewers.

Maria Rosaria Corbo

Reviewer 1 ·

Basic reporting

The article is well written, includes sufficient introduction and backgroud to demonstrate how the work fit into the broader field of knowledge. The structure of the article conforms the PeerJ policies. Figure are relevant to the content of the article and appropriately described.

Experimental design

The submission describes original primary research, methods are described with sufficient information to be reproducible by another investigator. The investigation has been conducted rigourosly although some limitations of the methods applied the also the authors underlined.

Validity of the findings

The data are robust but they have been considered within the limitations of the method applied. The conclusion are appropriately stated and connected to the original question investigated.

Reviewer 2 ·

Basic reporting

The authors affirm that the objective of this study is to develop a food composition database for 7 discretionary food categories of packaged GF products available on the Austrian market and determine their cost range with the goal to support individuals with celiac disease in their dietary choices. Nutrient composition, nutritional information and cost of foods were systematically obtained from 12 different supermarkets. The nutrition composition (macro and micronutrients) of 63 GF and 126 gluten-containing food was analyzed by using two nutrient composition databases in a stepwise approximation process.
As written, the manuscript has some serious problems:
• The authors did not analyze the nutritional composition of GF foods through direct chemical analysis, but they estimated their amount from labels. As stated by the same authors, these data are estimated without a real analysis of the food products.
• Very few products were analyzed (only 63)!!
• The collected (read) data are badly exposed. For each gluten-free product (table 1 and table 2), the comparison with the gluten-containing counterpart should be added. As presented, the paper appears very vague and confusing: it should be refocused stating clearly the aim and underlining the importance of the conducted research.
• The use of English is very poor.
The above mentioned problems make other minor details through the text of little relevance to be underlined.

Experimental design

The authors did not analyze the nutritional composition of GF foods through direct chemical analysis, but they only read labels.

Validity of the findings

Very few samples were analyzed.
The data on which the conclusions are based were not obtained by a real chemical analysis.

Additional comments

In my opinion the paper is not apt for publication.

Reviewer 3 ·

Basic reporting

In the present study, Missbach and coworkers report the some nutritional features and costs of the main gluten-free products available in Austria. A comparison with the related gluten-containing foods was also reported. The manuscript is of interest and the matter appropriate for the journal. 1. I feel the paper would benefit from careful checkup of the English used (e.g., L. 56-59; L. 64-68, L. 110-111; etc).

Experimental design

Several critical concerns of the materials and methods indicate that the manuscript in its current form is inadequate for publication.
There are some points that the authors should further discuss or clarify:
1. As stated by the Authors a limitation of this study is the method used to determine the nutritional composition of foods. In addition, the provenience of each food is not provided in the manuscript. This information could be added as supplementary materials.
2. The total number of GF and non-GF foods is dissimilar (63 vs 126) to have a good statistical evaluation of the data.

Validity of the findings

Gluten-free food database: to add merit to this publication an online interactive database for Austrian gluten-free foods should be performed.
Minor comments:
1. Please specify if the data in Figures and Tables are median or average.
2. Please added p value in fig. 2
3. Statistical analyses must be added in Table 1 and 2.
4. Please added P value in Table 3.
5. L. 302: the name of the journal should be written with the first letter capitalized
6. L. 309-310; 340-341; 352-353; 360-361; 366-368; 392-394; 395-397: the words of the title of the article should be not written with the first letter capitalized
7. L. 324; L. 329-330; 336; 339; 350-351; 362-365: please revise the name of the Journal
8. L. 400: Please complete the reference.

---

## Round 0.2 · accepted · Accept

· Academic Editor

Accept

Dear Authors,

Thank you for your revised submission. All the points raised by the reviewers were clarified therefore your paper is now acceptable for publication in PeerJ.

Best regards,
Maria Rosaria Corbo